**Data Availability Statement:** All relevant data are within the paper and its Supporting Information files.

**Funding:** We received no specific funding for this work.

# High global consumption of potentially inappropriate fixed dose combination antibiotics: Analysis of data from 75 countries

Barbara Bortone[1], Charlotte Jackson[2], Yingfen Hsia [2,3]*, Julia Bielicki[2,4], Nicola Magrini[5], Mike Sharland[2]

**1** Paediatric Infectious Diseases Division, Meyer Children's University Hospital, Florence, Italy, **2** St George's, University of London, Paediatric Infectious Diseases Research Group, Institute of Infection and Immunity, London, United Kingdom, **3** School of Pharmacy, Queen's University of Belfast, Belfast, United Kingdom, **4** Paediatric Pharmacology and Paediatric infectious Diseases, University of Basel Children's Hospital, Basel, Switzerland, **5** WHO, Expert Committee on the Selection and Use of Essential Medicines, Geneva, Switzerland

* Y.Hsia@qub.ac.uk

## Abstract

Antibiotic fixed dose combinations (FDCs) can have clinical advantages such as improving effectiveness and adherence to therapy. However, high use of potentially inappropriate FDCs has been reported, with implications for antimicrobial resistance (AMR) and toxicity. We used a pharmaceutical database, IQVIA-Multinational Integrated Data Analysis System (IQVIA-MIDAS®), to estimate sales of antibiotic FDCs from 75 countries in 2015. Antibiotic consumption was estimated using standard units (SU), defined by IQVIA as a single tablet, capsule, ampoule, vial or 5ml oral suspension. For each FDC antibiotic, the approval status was assessed by either registration with the United States Food and Drug Administration (US FDA) or inclusion on the World Health Organization (WHO) Essential Medicines List (EML). A total of 119 antibiotic FDCs were identified, contributing 16.7 x $10^9$ SU, equalling 22% of total antibiotic consumption in 2015. The most sold antibiotic FDCs were amoxicillin-clavulanic acid followed by trimethoprim/sulfamethoxazole and ampicillin/cloxacillin. The category with the highest consumption volume was aminopenicillin/β-lactamase inhibitor +/- other agents. The majority of antibiotic FDCs (92%; 110/119) were not approved by the US FDA. Of these, the most sold were ampicillin/cloxacillin, cefixime/ofloxacin and metronidazole/spiramycin. More than 80% (98/119) of FDC antibiotics were not compatible with the 2017 WHO EML. The countries with the highest numbers of FDC antibiotics were India (80/119), China (25/119) and Vietnam (19/119). There is high consumption of FDC antibiotics globally, particularly in middle-income countries. The majority of FDC antibiotic were not approved by either US FDA or WHO EML. International initiatives such as clear guidance from the WHO EML on which FDCs are not appropriate may help to regulate the manufacturing and sales of these antibiotics.

**Competing interests:** The authors have declared that no competing interests exist.

## Introduction

Global antibiotic consumption has been changing rapidly, increasing by 65% between 2000 and 2015, mainly in low- and middle-income countries (LMICs) [1,2]. Fixed dose combinations (FDCs) including one or more products with antibacterial activity have previously been noted as a concern, but there has been limited data on the scale of their use [3–6]. FDCs are defined by the World Health Organization (WHO) as "*A combination of two or more actives in a fixed ratio of doses. This term is used generically to mean a particular combination of actives irrespective of the formulation or brand. It may be administered as single entity products given concurrently or as a finished pharmaceutical product*" [7]. These products can have advantages such as improving treatment response compared to monotherapy, due to synergistic mechanisms (such as sulfamethoxazole/trimethoprim), or by increasing adherence to therapy.

FDCs are well-established in conditions such as tuberculosis, malaria and HIV treatment. However, the consumption of potentially clinically inappropriate antibiotic FDCs has been reported in some countries, raising concerns about the lack of proven efficacy, increasing toxicity or their potential effect on selecting for antimicrobial resistance (AMR) [4–6]. In 2017, the WHO Essential Medicines List (EML) Working Group classified antibiotics in the EML and EML for Children (EMLc) into three groups: Access, Watch, and Reserve (AWaRe classification). The Access group contains generally narrow spectrum antibiotics recommended as first and second choice for most common clinical infection syndromes. The Watch group contains broader spectrum antibiotic classes corresponding to the highest priority agents on the list of critically important antimicrobial drugs for human medicine. The Reserve group consists of last resort antibiotics for targeted use in multidrug resistant infections (Sharland et al., 2018). The new AWaRe classification is intended to be easy to apply to monitor antibiotic use and inform antibiotic stewardship. We therefore aimed to quantify the global consumption of antibiotic FDCs and to describe the types of combinations used. We also identified their approval status with the US Food and Drug Administration (FDA) and compatibility with the WHO's 2017 revision of the Essential Medicines List (EML).

## Methods

### Data

This was an ecological study to assess the total sale of antibiotic FDCs. We estimated global antibiotic FDCs sales in 2015 using the IQVIA-Multinational Integrated Data Analysis System (IQVIA-MIDAS®) database. IQVIA-MIDAS is a commercial database containing data from pharmacy retails sales throughout the supply chains, including overall antibiotic volume sold to retailers and hospital pharmacies by wholesalers. The proportion of wholesalers contributing data to IQVIA-MIDAS varies between represented countries and IQVIA adjusts the reported data based on the market share of participating wholesalers, to provide estimates of total sales in the sectors represented in each country. Our dataset contains annual pharmaceutical sales data for 75 countries/regions. Central America (Costa Rica, El Salvador, Guatemala, Honduras, Nicaragua, and Panama) and Francophone West Africa (Benin, Burkina Faso, Cameroon, Chad, Côte d'Ivoire, Republic of Congo, Guinea, Mali, Niger, Senegal, and Togo) are defined as two regions with aggregated sales for these countries in the database [8].

Annual sales of each antibiotic (including information on component antibiotics, formulation, trade name and manufacturer) are recorded for each country. Antibiotic sales were expressed in standard units (SU), with 1 SU defined by IQVIA as one tablet, capsule or ampoule/vial or 5ml oral suspension. We did not include anti-tuberculosis drugs, antiviral drugs, and antifungal drugs in our analyses.

## Analysis

We defined antibiotic FDCs as medications consisting of at least one Anatomical Therapeutic Chemical (ATC) code J01 systemic antibiotic with another J01 antibiotic or a β-lactamase inhibitor or a 5-nitroimidazole with or without any other medication (e.g probiotics, non-steroidal anti-inflammatory drugs). Antibiotic FDCs were aggregated in categories based on the WHO ATC Classification System [9]. Penicillins were classified as β-lactamase sensitive penicillins, β-lactamase resistant penicillins, aminopenicillins, ureidopenicillins and carboxypenicillins. Ureidopenicillins and carboxypenicillin were grouped as antipseudomonal penicillins [10]. The total annual sales of each antibiotic FDC sold was recorded by formulation type. For each antibiotic FDC, we summed the total consumption and reported the number of countries where it was sold. Total consumption for each antibiotic FDC was quantified by country income. The World Bank categories were used to classify countries as high income and low/middle income countries (HICs and LMICs) [11].

**Approval status of antibiotic FDCs.** Each of the antibiotic FDCs was categorised as approved or not approved by the US FDA by searching the FDA website [12]. For each country, the number and volume of antibiotic FDCs not approved by the FDA was expressed as a percentage of total antibiotic sales and total antibiotic FDCs.

**Classification of antibiotic FDCs according to WHO criteria.** Antibiotic FDCs were also assessed based on the WHO 2017 revision of EML [13]. EML compatible antibiotic FDCs were grouped into Access, Watch, and Reserve using the WHO AWaRe classification.

**Antibiotic FDCs with two critically important antibiotics.** The components of the antibiotic FDCs were also matched to the WHO's "critically important antibiotics" (CIA) classification [14] to identify antibiotic FDCs in which both the components belonged to the "highest priority critically important antibiotics" group.

## Results

A total of 74.4 x $10^9$ SU of antibiotic agents was sold in 2015. Amongst these, 119 different antibiotic FDCs were identified (S1 Table), contributing to 16.7 x $10^9$ SU, 22.5% of all antibiotic sales. The highest number of antibiotic FDCs were in the category of aminopenicillin/β-lactamase resistant penicillin (Table 1). The category with the greatest sales was that of aminopenicillin /β-lactamase inhibitor +/- other agents (Table 1).

Approximately one in six (20/119) antibiotic FDCs included probiotics as additional agents; probiotics were mostly *Lactobacillus acidophilus* and *Bacillus coagulans*. The most sold was cefixime/dicloxacillin/*Lactobacillus acidophilus*. The country with the highest number of antibiotic FDCs sold was India (80), followed by China (25) and Vietnam (19) (Fig 1). Overall, the most sold FDC by volume was amoxicillin/clavulanic acid (8.38 x $10^9$ SU), followed by sulfamethoxazole/trimethoprim (3.61 x $10^9$ SU) and ampicillin/cloxacillin (0.95 x $10^9$ SU) (Table 2 and S1 Table). Amoxicillin/clavulanate and sulfamethoxazole/trimethoprim were sold by all 75 countries; other widely sold antibiotic FDCs were piperacillin/tazobactam (66) and ampicillin/sulbactam (49) (Table 2).

Compared to other countries, India accounted for the greatest sales volume of amoxicillin/clavulanic acid, sulfamethoxazole/trimethoprim, and ampicillin/cloxacillin. There were clear country level differences in sales of FDCs, with the highest sales of piperacillin/tazobactam in Australia, metronidazole/spiramycin in France and cefoperazone/sulbactam in China.

### FDA approval

Only 9/119 antibiotic FDCs were approved by the US FDA (7.5%) (Table 3). Of the 110 antibiotic FDCs not authorized by US FDA (92.5%), the highest numbers of combinations and the

**Table 1. Categories of antibiotic Fixed Dose Combinations (FDCs).**

| FDC types | Standard Unite sold | Number of FDCs |
|---|---|---|
| Aminopenicillin /β-lactamase inhibitor //- other agents | $8.60 \times 10^9$ | 8 |
| Sulphonamides/trimethoprim+/- other agents | $3.62 \times 10^9$ | 9 |
| Aminopenicillin / β-lactamase resistant penicillin +/- other agents | $1.54 \times 10^9$ | 21 |
| Antipseudomonal penicillin /β-lactamase inhibitor | $0.95 \times 10^9$ | 4 |
| 3rd-4th-5th gen. cephalosporins /β-lactamase inhibitor +/- other agents | $0.55 \times 10^9$ | 15 |
| Cephalosporins / fluoroquinolones | $0.40 \times 10^9$ | 6 |
| 1st-2nd gen. cephalosporins / β-lactamase inhibitor +/- other agents | $0.26 \times 10^9$ | 8 |
| Macrolide/ 5-nitroimidazole | $0.24 \times 10^9$ | 3 |
| Macrolide/cephalosporin+/-other agents | $0.21 \times 10^9$ | 3 |
| Cephalosporin/ β-lactamase resistant penicillin +/- other agents | $0.10 \times 10^9$ | 7 |
| Cephalosporin/trimethoprim | $0.09 \times 10^9$ | 2 |
| Cephalosporin/oxazolidinone | $0.04 \times 10^9$ | 2 |
| Fluoroquinolone/ 5-nitroimidazole | $0.04 \times 10^9$ | 8 |
| Macrolide / fluoroquinolone +/- other agents | $0.04 \times 10^9$ | 2 |
| Cephalosporin/5-nitroimidazole | $0.03 \times 10^9$ | 1 |
| Other combinations | $0.01 \times 10^9$ | 20 |

largest sales volumes were in the categories containing aminopenicillin+β-lactamase resistant penicillins +/- other agents (probiotics, lactic acid, serrapeptase) (19) and that including 3rd-4th-5th generation cephalosporins/β-lactamase resistant penicillins +/- other agents (13) (S2 Table).

Overall, 53/75 (70.7%) countries sold at least one FDC not approved by the US FDA (Fig 1); 31/53 countries (58.4%) were LMIC. The highest number of "not FDA-approved" antibiotic FDCs were sold in India, China, Francophone West Africa and Vietnam (Fig 1). India and Francophone West Africa had the highest percentage of antibiotic FDCs not approved by the US FDA (93.8% of total antibiotic FDCs). Ampicillin/cloxacillin was the highest sold antibiotic FDC which was not FDA-approved, followed by cefixime/ofloxacin, and metronidazole/spiramycin (Table 4). A total of $3.81 \times 10^9$ SU of antibiotic FDCs not approved by the US FDA were sold, corresponding to 22.7% of the antibiotic FDCs SU sold and to 5.1% of the total antibiotic consumption, with considerable variation across countries (S3 Table).

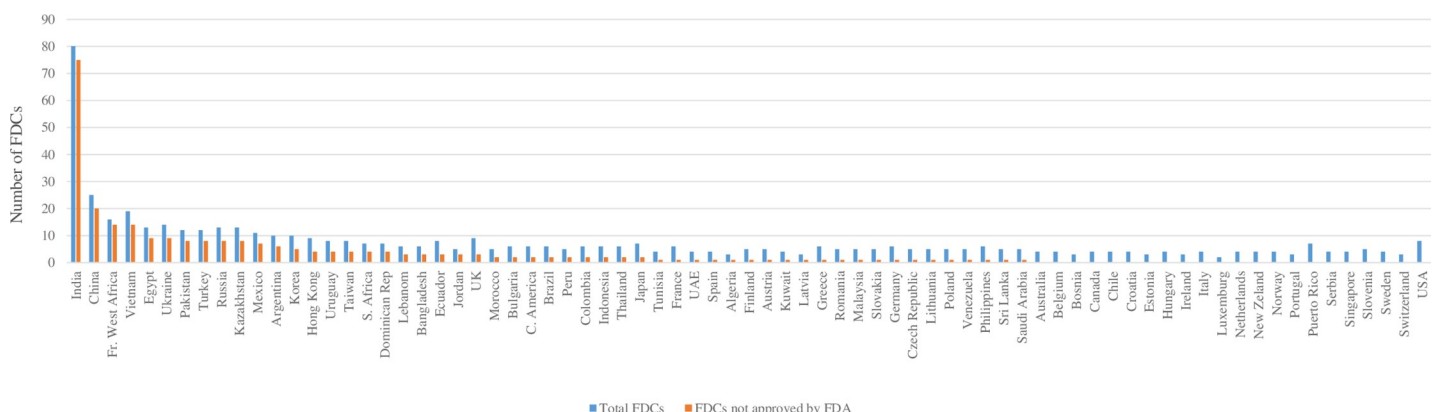

**Fig 1. Number of fixed dose combination (FDC) antibiotic sold by country.**

**Table 2. Top 10 list of most sold antibiotic FDCs and the number of countries they are sold in.**

| FDC | Standard Unite sold | Number of countries sold in |
|---|---|---|
| amoxicillin/clavulanic acid | $8.38 \times 10^9$ | 75 |
| sulfamethoxazole/trimethoprim | $3.61 \times 10^9$ | 75 |
| ampicillin/cloxacillin | $0.95 \times 10^9$ | 13 |
| piperacillin/tazobactam | $0.79 \times 10^9$ | 66 |
| cefixime/ofloxacin | $0.31 \times 10^9$ | 1 |
| metronidazole/spiramycin | $0.24 \times 10^9$ | 17 |
| cefpodoxime proxetil/clavulanic acid | $0.23 \times 10^9$ | 3 |
| amoxicillin/flucloxacillin | $0.19 \times 10^9$ | 7 |
| azithromycin/cefixime | $0.17 \times 10^9$ | 2 |
| cefoperazone/sulbactam | $0.16 \times 10^9$ | 28 |

## Compatibility of antibiotic FDCs with the WHO EML

Among the 119 antibiotic FDCs, 98 (82.3%) were not compatible with the 2017 WHO EML (S4 Table). Among the 21 WHO-compatible antibiotic FDCs (S3 Table), 2 belonged to the "Access" category (amoxicillin/clavulanic acid and sulfamethoxazole/trimethoprim), 16 were compatible with the "Watch" group and 3 with the "Reserve" group (S4 Table). Ten antibiotic FDCs included two highest priority critically important antibiotics (S5 Table). Of these, the largest consumption volumes were reported for cefixime/ofloxacin and azithromycin/cefixime; none of these 10 antibiotic FDCs were listed on the EML or approved by the US FDA.

## Discussion

We have analysed sales data on antibiotic fixed dose combinations from 75 countries, adding to previous studies which used data from single countries or regions [4–6]. Overall, antibiotic FDCs represent a substantial proportion of all systemic antibiotic consumption, accounting for about 20% of systemic antibiotic SU sold in 2015. The majority of the antibiotic FDCs identified were not compatible with the EML and were not approved by the US FDA, and several antibiotic FDCs contained two agents classified by the WHO as highest priority critically important antibiotics.

Globally, the most sold antibiotic FDCs were amoxicillin/clavulanate and sulfamethoxazole/trimethoprim, which were sold in all countries and accounted for about 50% and 20% of the total antibiotic FDCs consumption, respectively. This is consistent with these drugs being recommended for treatment of common infectious conditions (such as acute otitis media, community acquired pneumonia and urinary tract infections) and with their inclusion on the

**Table 3. Antibiotic FDCs authorized by FDA and the number of countries they are sold in.**

| FDCs | Standard Unite sold globally | Number of countries sold in |
|---|---|---|
| amoxicillin/clavulanic acid | $8.38 \times 10^9$ | 75 |
| sulfamethoxazole/trimethoprim | $3.61 \times 10^9$ | 75 |
| piperacillin/tazobactam | $0.79 \times 10^9$ | 66 |
| ampicillin/sulbactam | $0.12 \times 10^9$ | 49 |
| clavulanic acid/ticarcillin | $0.02 \times 10^9$ | 19 |
| phenazopyridine/sulfamethoxazole/trimethoprim | $0.02 \times 10^8$ | 1 |
| ceftolozane/tazobactam | $0.01 \times 10^7$ | 9 |
| avibactam/ceftazidime | $0.07 \times 10^6$ | 2 |
| dalfopristin/quinupristin | $0.02 \times 10^6$ | 3 |

**Table 4. Antibiotic FDCs sold in the highest volumes in 2015 and not FDA approved FDCs and number of countries sold in.**

| FDC | Standard Unite globally sold | Number of countries sold in |
|---|---|---|
| ampicillin/cloxacillin | $0.95 \times 10^9$ | 13 |
| cefixime/ofloxacin | $0.31 \times 10^9$ | 1 |
| metronidazole/spiramycin | $0.24 \times 10^9$ | 17 |
| cefpodoxime proxetil/clavulanic acid | $0.22 \times 10^9$ | 3 |
| amoxicillin/flucloxacillin | $0.19 \times 10^9$ | 7 |
| azithromycin/cefixime | $0.17 \times 10^9$ | 2 |
| cefoperazone/sulbactam | $0.16 \times 10^9$ | 28 |
| amoxicillin/cloxacillin | $0.16 \times 10^9$ | 4 |
| cefixime/clavulanic acid | $0.12 \times 10^9$ | 2 |
| cefalexin/trimethoprim | $0.09 \times 10^9$ | 1 |

EML as Access antibiotics [13]. However, there were many other widely used antibiotic FDCs, particularly from middle income countries, that were not on the EML. The category including the highest number of types of antibiotic FDCs was that of aminopenicillin/ β-lactamase resistant penicillins with or without probiotics followed by 3rd-4th-5th generation cephalosporins plus β-lactamase inhibitors with or without probiotics.

Defining clinically "inappropriate" antibiotic FDCs is complex and beyond the scope of this study. Previous studies have used as a standard of appropriateness the authorization status of antibiotic FDCs by the FDA, UK Medicines and Healthcare products Regulatory Agency (MHRA) and European Medicines Agency (EMA) and local drug regulatory organisations [4–6]. We only used the US FDA to evaluate approval status for marketed FDC antibiotics in this study, as US FDA regulation is a widely accepted benchmark for drug approvals globally, while the EMA has both centralised and devolved local regulations for drug approval processes.

However, countries with different health priorities to the US may investigate and approve drugs (including antibiotic FDCs) which have not been approved by the US FDA. Therefore, in some countries, the use of antibiotic FDCs not approved by the US FDA might be clinically appropriate (e.g. those including aminopenicillin/β-lactamase inhibitor, anti-pseudomonas penicillin / β-lactamase inhibitor or a third-fourth-fifth generation cephalosporins/β-lactamase inhibitor and the wider group of sulphonamides/trimethoprim) [15]. Some of these antibiotic FDCs are part of the Watch and Reserve groups of the EML [13], therefore their consumption should be further monitored, investigated and potentially limited to selected indications.

On the other hand, for other combinations there appears to be no potential clinical additional benefit compared to the individual drugs formulated alone, such as those including two drugs belonging to the same class (cefpodoxime proxetile/cefixime, ampicillin/sultamicillin). There is limited evidence supporting the use of the three antibiotic FDCs accounting for the greatest consumption (ampicillin/cloxacillin, cefixime/ofloxacin and metronidazole/spiramycin). The clinical utility of using ampicillin/cloxacillin has been questioned [16–18], as the two antibiotics have overlapping spectra of activity and medical indications that require both these antibiotics as an empiric therapy are not common [16]. Cefixime/ofloxacin was only sold in India where the oral formulation was approved in 2010 for treating typhoid fever [19] in response to the increasing prevalence of *Salmonella typhi* resistant to quinolones [20]. Two randomized clinical trials are ongoing to compare cefixime/ofloxacin to ofloxacin for the therapy of typhoid fever, and may provide evidence about clinical effectiveness and safety of this combination [21,22]. Metronidazole/spiramycin (indicated for treatment of periodontitis) [23] was sold mainly in France and Vietnam. It has been reported that these two antibiotics

have shown synergism in vitro against oral cavity bacteria and their efficacy was similar to amoxicillin-clavulanate but also to metronidazole monotherapy [24]. For this reason, it has been recommended in patients with history of hypersensitivity to amoxicillin/clavulanate.

## Concerns regarding clinical efficacy

The approval of antibiotic FDCs may not always require evidence of efficacy and toxicity [7]. Evidence of clinical efficacy and safety of the single components or studies on the combination of the two agents can be sufficient, even if not in a fixed dose ratio [7]. A review of 22 different Indian brands of FDCs containing azithromycin with ofloxacin or cefixime reported that while the formulations contained an adequate dose of fluoroquinolone or cephalosporin, the dose of azithromycin was only 250 mg/tablet (one fifth of the WHO-recommended dose for a 60 kg individual (1200 mg; 10 mg/kg per day)) [25]. In India, dosing of ceftriaxone/vancomycin is recommended twice a day [26]. This dosing frequency is widely accepted for ceftriaxone [27] but administration of vancomycin is usually recommended every 6–8 hours [28].

## Concerns regarding safety

There are limited data comparing toxicity resulting from antibiotic FDCs versus monotherapy for conditions other than tuberculosis. However, antibiotic FDCs (particularly those classified as irrational based on the 2012 WHO Essential Drug Model list) have been associated with about 20% of the total adverse drug reactions as a result of antibiotics (26). The combination most commonly associated with adverse events was Ofloxacin/Ornidazole, in terms of cutaneous manifestations [29]. Moreover, some antibiotic FDCs not approved by the FDA include narrow therapeutic index antibiotics such as glycopeptides and aminoglycosides (ceftriaxone/vancomycin, amikacin/cefepime).

## Concerns regarding AMR

Gautam et al suggested a possible role of the injudicious use of antibiotic FDCs in the emergence of ciprofloxacin-resistant *Salmonella typhi* strains in India [30]. However, this is a complex area as the development of antibiotic FDCs may in part be a response to high AMR rates (as with the licensing in India of cefixime/ofloxacin for typhoid fever) [31,32]. Antibiotic FDCs may contribute to AMR by exposing bacteria to sub-therapeutic concentrations of one or both antibiotic components since sub-therapeutic concentrations can exert non-lethal selective pressures [31]. This selection of resistant strains may further contribute to a reduction in clinical effectiveness.

## Country specific FDC antibiotic use

Antibiotic FDCs not approved by the US FDA were most commonly sold in middle-income countries, particularly in India, where the sale of US FDA unapproved FDCs accounts for around 50% of antibiotic FDCs and about 16% of overall antibiotics, considerably higher than many other countries. These differences between countries in terms of number and consumption of antibiotic FDCs not approved by the US FDA can be related to several factors, including differences in government regulation [6]. For example, in India, many antibiotic FDCs have historically been sold without authorisation from the Central Drugs Standard Control Organisation (CDSCO), the national regulator [33]. Since 2014/5, the CDSCO has been providing reports of the rationality of FDCs, thus leading the government to ban their manufacture, sale and distribution [4,34]. Among these products, several antibiotics FDCs are listed [35,36].

The many factors which contribute to national differences in consumption of antibiotic FDCs include different prevalence of infectious diseases, access to medical facilities and appropriate diagnostic procedures, accessibility of essential drugs [17], inappropriate prescribing practices [37] and over the counter and non-prescription supply of antimicrobials [1]. Contributing factors for non-prescription antimicrobial supply are poor national medicines regulations, limited availability of qualified pharmacists and commercial pressure on pharmacy staff, consumer demand and lack of awareness of AMR [38]. Two recent studies in China and Vietnam have shown that patients were able to purchase non-prescription antibiotics through various sources (e.g. online pharmacy, illegal drug suppliers) [39,40]. The rationale for individuals using and healthcare workers providing FDC antibiotics should be further explored. Studies are also required to better evaluate their efficacy, effectiveness, safety and potential to accelerate the development of AMR.

## Strengths and limitations

To our knowledge, this is the first comprehensive study to assess FDC antibiotic sales at the global level. However, several limitations need to be addressed. The IQVIA database contains aggregate, country-level data without information about indications of therapy, consumers' age, or setting (hospital, community). The sales data were not adjusted for population size, which partly explains the apparently large contribution of highly populated countries such as India. The IQVIA database is not globally representative (for example, only two low income countries contribute data; these are both part of Francophone West Africa); however, it does provide valuable information on the scale of antibiotic use in the included countries [8]. The IQVIA company estimates coverage for each country internally and applies correction factors to estimate total sales in the sectors covered in each country. Data are also validated against alternative data sources [41]. Despite this uncertainty about the precise volume of FDC antibiotics sold and the validity of comparisons between countries, our analyses support the conclusion that antibiotic FDCs are sold in large volumes in many countries. Finally, it is not possible to distinguish sales of prescription and non-prescription antibiotics in this database.

As discussed above, US FDA authorization cannot be considered as the gold standard measure of a drug's appropriateness. We use this for illustrative purposes and not to formally assess the appropriateness of particular antibiotic FDCs in a given context; we acknowledge that individual countries may have valid reasons to approve antibiotic FDCs not approved (or not assessed) by the US FDA.

## Conclusions

A high consumption of FDC antibiotic globally was observed, particularly in middle-income countries. Antibiotic FDCs account for a substantial proportion of antibiotic consumption and some of these FDCs may not be clinically appropriate, depending on the context. Considering the reported concerns in terms of efficacy, toxicity and AMR, the rationale for using antibiotic FDCs should be explored further and where required studies conducted to assess their effectiveness, safety and potential to accelerate the development of AMR. International initiatives may be needed to regulate the manufacturing and sale of these medications while maintaining access to essential antibiotics, with particular attention to LMICs. Central guidance from the WHO EML could be helpful to assist in determining the clearly inappropriate FDCs and discouraging their manufacture and sale.

## Supporting information

**S1 Table. Antibiotic FDCs in 2015 sorted by SU globally sold.**
(DOCX)

**S2 Table. Antibiotic FDC categories not approved by US FDA.**
(DOCX)

**S3 Table. Total antibiotic sales and FDCs not approved by the FDA as a percentage of total sales and of total antibiotic FDC sales per each country selling at least one antibiotic FDC not approved by FDA, sorted by the highest not approved FDCs/ antibiotics.**
(DOCX)

**S4 Table. Antibiotic FDCs compatible with WHO EML and AWaRe classes.**
(DOCX)

**S5 Table. Antibiotic FDCs combinations including two highest critically important antibiotics.**
(DOCX)

## Acknowledgments

We are grateful to Peter Stephens for his helpful technical input on the IQVIA-MIDAS data. The findings and suggestions in this paper are based on data obtained under licence from IQVIA information services: IQVIA-MIDAS sale data; all rights reserved. The statement and suggestions are not necessarily those of IQVIA Health Incorporated or any of its affiliated or subsidiary entities.

## Author Contributions

**Conceptualization:** Charlotte Jackson, Yingfen Hsia, Mike Sharland.

**Data curation:** Barbara Bortone, Charlotte Jackson, Yingfen Hsia, Mike Sharland.

**Formal analysis:** Barbara Bortone, Charlotte Jackson.

**Investigation:** Barbara Bortone, Charlotte Jackson, Julia Bielicki, Nicola Magrini, Mike Sharland.

**Methodology:** Julia Bielicki, Nicola Magrini, Mike Sharland.

**Supervision:** Charlotte Jackson, Yingfen Hsia, Nicola Magrini, Mike Sharland.

**Validation:** Barbara Bortone.

**Visualization:** Julia Bielicki.

**Writing – original draft:** Barbara Bortone, Charlotte Jackson, Yingfen Hsia, Julia Bielicki, Nicola Magrini, Mike Sharland.

**Writing – review & editing:** Barbara Bortone, Charlotte Jackson, Yingfen Hsia, Julia Bielicki, Nicola Magrini, Mike Sharland.

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
