## [Decision Letter · Decision Letter 0]

6 Mar 2020

PONE-D-20-01816

High global consumption of potentially inappropriate fixed dose combination antibiotics: analysis of data from 75 countries

PLOS ONE

Dear Dr Hsia,

Thank you for submitting your manuscript to PLOS ONE. After careful consideration, we feel that it has merit but does not fully meet PLOS ONE’s publication criteria as it currently stands. Therefore, we invite you to submit a revised version of the manuscript that addresses the points raised during the review process.

 The reviewers have raised number of questions on methodology and results as well as discussion. All the points raised by reviewers need to be addressed.

We would appreciate receiving your revised manuscript by Apr 20 2020 11:59PM. To enhance the reproducibility of your results, we recommend that if applicable you deposit your laboratory protocols in protocols.io, where a protocol can be assigned its own identifier (DOI) such that it can be cited independently in the future. For instructions see: http://journals.plos.org/plosone/s/submission-guidelines#loc-laboratory-protocols

We look forward to receiving your revised manuscript.

Kind regards,

Iddya Karunasagar

Academic Editor

PLOS ONE

Journal Requirements:

Additional Editor Comments (if provided):

The reviewers have raised a number of questions regarding methodology, presentation of results and discussion. Please revise the manuscript addressing all questions.

Reviewers' comments:

Reviewer's Responses to Questions

**Comments to the Author**

1. Is the manuscript technically sound, and do the data support the conclusions?

Reviewer #1: Yes

Reviewer #2: Yes

2. Has the statistical analysis been performed appropriately and rigorously? 

Reviewer #1: Yes

Reviewer #2: Yes

3. Have the authors made all data underlying the findings in their manuscript fully available?

Reviewer #1: Yes

Reviewer #2: Yes

4. Is the manuscript presented in an intelligible fashion and written in standard English?

Reviewer #1: Yes

Reviewer #2: Yes

5. Review Comments to the Author

Reviewer #1: I read the manuscript with interest. I appreciate the authors for focusing the study on the issue of use of fixed dose combinations. The manuscript presents consumption data of fixed dose combinations from 75 countries with limited data available.

Reviewer #2: The manuscript by Hsia Y and colleagues report on the high consumption of fixed dose combination (FDC) antibiotics globally from 75 countries by utilizing a pharmaceutical database, IQVIA-Multinational Integrated Data Analysis System (IQVIA- MIDAS). They conclude that India, China and Vietnam had the highest sales of FDC antibiotics, majority of which did not have approval by the United States Food and Drug Administration (U.S. FDA) and the World Health Organization (WHO) Essential Medicines List (EML).

Introduction:

a. Expand on the introduction to include details about 2017 WHO Essential Medicine List and National List of Essential Meds (NLEMs).

b. Mention and define WHO classification of antibiotics with citation of appropriate reference [ref 13].

Methods:

a. ?Ecological study

b. Describe antibiotic sales data in more detail as captured in the IQVIA database. For example, mention the sources of the sales audit were collected (besides pharmaceutical industries, ?distributers, ?sub-stockists, ?retailers and hospitals, and ?dispensing physicians, etc.

c. What is the percentage of national antibiotic sales for India, China and Vietnam captured in the IQVIA database?

d. Are the sales volume reported monthly? Please clarify.

e. How many investigators examined the IQVIA database? Was any discrepancy observed?

f. Mention exclusion criteria if any?anti-TB drugs, antivirals, antifungals, etc

g. Please indicate if your study adhered to the Strengthening the Reporting of Observational Studies in Epidemiology – Antimicrobial Studies (STROBE-AMS) for reporting..if Yes, please state and cite reference Tacconelli E, et al. BMJ Open 2016;6:e010134.

Results:

a. Mention sales of unapproved single-drug formulations (SDFs) if data captured in the IQVIA database.

Discussion:

a. The authors must discuss their Recommendations in more detail in a separate paragraph

- Government ban, Regulation for MNCs to justify sale of unapproved products. Public awareness of approved FDC. (Holloway KA, et al. PLoS ONE 2016;11:e0152020)

Mention if there is data on why clinicians choose unapproved antibiotic FDCs in countries such as India and China? Role of MNCs, conflict of interest, etc.

b. Study Limitations:

- ?possibility of error in sampling methodology.

- Can the IQVIA database distinguish sales of prescription vs. non-prescription antibiotics. This should be mentioned in the limitations.

References:

a. Reference 4 – cite year, volume, page no for Br J Clin Pharmacol.

b. Reference 36 – newspaper article, cite online link if available

c. Reference 38 – cite volume, page no

d. Please follow journal instructions for formatting of references

6. PLOS authors have the option to publish the peer review history of their article (what does this mean?). If published, this will include your full peer review and any attached files.

Reviewer #1: No

Reviewer #2: No

---

## [Author Response · Author response to Decision Letter 0]

1 Oct 2020

Reviewer #1: I read the manuscript with interest. I appreciate the authors for focusing the study on the issue of use of fixed dose combinations. The manuscript presents consumption data of fixed dose combinations from 75 countries with limited data available.

Response: Thank you. 

Reviewer #2: The manuscript by Hsia Y and colleagues report on the high consumption of fixed dose combination (FDC) antibiotics globally from 75 countries by utilizing a pharmaceutical database, IQVIA-Multinational Integrated Data Analysis System (IQVIA- MIDAS). They conclude that India, China and Vietnam had the highest sales of FDC antibiotics, majority of which did not have approval by the United States Food and Drug Administration (U.S. FDA) and the World Health Organization (WHO) Essential Medicines List (EML).

Introduction:

a. Expand on the introduction to include details about 2017 WHO Essential Medicine List and National List of Essential Meds (NLEMs).

Response: Thank you for the comment. We have expanded and included details about 2017 WHO Essential Medicine List (EML) in the Introduction to place the rationale for our study in a global context. 

‘In 2017, the WHO Essential Medicines List (EML) Working Group classified antibiotics in the EML and EML for Children (EMLc) into three groups: Access, Watch, and Reserve (AWaRe classification). The Access group contains generally narrow spectrum antibiotics recommended as first and second choice for most common clinical infection syndromes. The Watch group contains broader spectrum antibiotic classes corresponding to the highest priority agents on the list of critically important antimicrobial drugs for human medicine. The Reserve group consists of last resort antibiotics for targeted use in multidrug resistant infections (Sharland et al., 2018). The new AWaRe classification is intended to be easy to apply to monitor antibiotic use and inform antibiotic stewardship.’ 

The primary purpose of the National List of Essential Medicines of India (NLEM) aims to promote rational use of medicines in India. Our study aimed to characterise global FDC antibiotics sales, we have not described guidance for individual countries (such as the NLEM of India) in our manuscript. [1] 

Methods:

a. ?Ecological study

Response: This is an ecological study and we have added this to the revised manuscript.

b. Describe antibiotic sales data in more detail as captured in the IQVIA database. For example, mention the sources of the sales audit were collected (besides pharmaceutical industries, ?distributers, ?sub-stockists, ?retailers and hospitals, and ?dispensing physicians, etc.

Response: Thank you for raising these comments. IQVIA-MIDAS is a commercial database. The data contained in the database were obtained from pharmacy retail sales throughout the supply chain, including overall antibiotic volume sold to retails and hospital pharmacies by wholesalers. IQVIA data were not contributed from all wholesalers in represented countries, adjustments had been made by the providers of the database to estimate total sales based on the market share of participating wholesalers. We don’t now know how IQVIA company estimate country-level coverage as those information are not publicly available. 

We have added more detailed on data captured in the IQVIA-MIDAS database in the revised manuscript. 

c. What is the percentage of national antibiotic sales for India, China and Vietnam captured in the IQVIA database?

Response: Unfortunately this information is not publicly available; however, the database as supplied includes adjustments for country-specific coverage as estimated by IQVIA to “scale up” to the total sales in the sector(s) covered in that country. The IQVIA database are internally validated against alternative source of sale data. We acknowledge this limitation and now include it in the Discussion section. 

d. Are the sales volume reported monthly? Please clarify.

Response: The database contains only annual sales volume and we have clarified this in the Methods section. 

e. How many investigators examined the IQVIA database? Was any discrepancy observed?

Response: We have three investigators (BB, CJ, YH) checked and compare classification of products as FDC antibiotics. There was no discrepancy was observed in our dataset. 

f. Mention exclusion criteria if any?anti-TB drugs, antivirals, antifungals, etc

Response: We did not include anti-TB drugs, antiviral drugs, and antifungals in our database. We have added the exclusion criteria in our revised manuscript. 

g. Please indicate if your study adhered to the Strengthening the Reporting of Observational Studies in Epidemiology – Antimicrobial Studies (STROBE-AMS) for reporting..if Yes, please state and cite reference Tacconelli E, et al. BMJ Open 2016;6:e010134.

Response: Thank you for the comment. We agree with reviewer that we should adhere to the STROBE-AMS. This statement is aimed at ‘epidemiologic studies focused on the link between antimicrobial resistant bacteria and antibiotic usage’. Our study only investigate the antibiotic usage so many items on the checklist are not relevant to our study. We did not adhere to all items on the checklist but we included a general on STROBE checklist when we conducted this study. 

Results:

a. Mention sales of unapproved single-drug formulations (SDFs) if data captured in the IQVIA database.

Response: We agree with the reviewer that it is also important to assess sales of unapproved single-drug formulations. However, this is the beyond the scope of this study. We appreciate the reviewer’s valuable comment and will consider further exploring the sales of unapproved single-drug formulations in future work. 

Discussion:

a. The authors must discuss their Recommendations in more detail in a separate paragraph

- Government ban, Regulation for MNCs to justify sale of unapproved products. Public awareness of approved FDC. (Holloway KA, et al. PLoS ONE 2016;11:e0152020)

Response: Thank you for the comments and we appreciate the useful reference. We agree with reviewer’s point of view. 

FDC antibiotic is a complex issue. Determining the evidence for the use of FDC antibiotics is the first step for policy makers to strengthen regulations for manufacturing these drugs. However, there is currently a lack of summarised international evidence to support the use of FDC antibiotics. We contacted a literature review for ampicillin-cloxacillin with the primary objective of summarising available data on the safety and efficacy of this FDC antibiotic. Our review highlights difficulties in accessing some potentially informative literature, particularly studies published in non-English language journals (paper is currently under review). We believe the best practical recommendation is the central guidance from the organization such as WHO could be helpful to assist to determine the clearly inappropriate FDC use. We already discuss this recommendation in Discussion section. 

-Mention if there is data on why clinicians choose unapproved antibiotic FDCs in countries such as India and China? Role of MNCs, conflict of interest, etc.

Response: We are not clear if reviewer refers to ‘quantitative data’ for this comment. If so, we are not aware any quantitative data to explore why clinicians choose unapproved antibiotic FDCs in India and China. However, there is a recent qualitive study conducted in Vietnam to interview 16 suppliers on their knowledge of antibiotics and antibiotic resistance: 1 hospital pharmacy; 1 commune health centre; 8 private dispensaries; 1 traditional medicine centre; 2 private clinics; 3 other types of selling antibiotics. This study highlights that patients were able to purchase antibiotics through different suppliers. The habit of using antibiotics driven not only by clinicians’ prescribing behaviour also suppliers and patients (‘customers’). [2] As we discussed above, the more practical recommendation should be the central guidance from WHO EML department. 

b. Study Limitations:

- ?possibility of error in sampling methodology.

Response: We acknowledge that the data provided to IQVIA-MIDAS may not be fully representative of antibiotic consumption in each contributing country. The IQVIA company does estimate coverage for each country internally and applies correction factors to estimate total sales in the sectors covered in each country. Data are also validated against alternative data sources. [3]

The database do not represent precise estimates of antibiotic FDC sales (they are likely to underestimate sales due to incomplete coverage) and that there are difficulties in comparing between countries. Despite this, our analysis indicates that FDCs are widely used throughout HICs and LMICs and we consider our conclusion that much of this use may be inappropriate and regulation should be explored is valid, even if we have not fully quantified sales. We acknowledge this and several other caveats which we all addressed in the Discussion. 

‘The sales data were not adjusted for population size, which partly explains the apparently large contribution of highly populated countries such as India. The IQVIA database is not globally representative (for example, only two low income countries contribute data; these are both part of Francophone West Africa); however, it does provide valuable information on the scale of antibiotic use in the included countries. The IQVIA company estimates coverage for each country internally and applies correction factors to estimate total sales in the sectors covered in each country. Data are also validated against alternative data sources’.

- Can the IQVIA database distinguish sales of prescription vs. non-prescription antibiotics. This should be mentioned in the limitations.

Response: We agree that this would be an important distinction, but unfortunately it is not possible to use the IQVIA-MIDAS database to distinguish sales of prescription vs non-prescription antibiotics. We acknowledge this limitation and have addressed it in the Discussion section : ‘Finally, it is not possible to distinguish sales of prescription and non-prescription antibiotics in this database’ 

References:

a. Reference 4 – cite year, volume, page no for Br J Clin Pharmacol.

b. Reference 36 – newspaper article, cite online link if available

c. Reference 38 – cite volume, page no

d. Please follow journal instructions for formatting of references

Response: Thank you for noting these errors, we have now corrected these references and follow the journal instructions for formatting the references. 

References

1. National List of Essential Medicines of India 2011. Avalable : https://pharmaceuticals.gov.in/sites/default/files/NLEM.pdf. [Accessed 10 Oct 2020]

2. Nguyen HH, Ho DP, Vu TLH, Tran KT, Tran TD, Nguyen TKC, van Doorn HR, Nadjm B, Kinsman J, Wertheim H. "I can make more from selling medicine when breaking the rules" - understanding the antibiotic supply network in a rural community in Viet Nam. BMC Public Health. 2019 Nov 26;19(1):1560.

3. IQVIA, 2018. ACTS: IQVIA Quality assurance. Available: https://www. iqvia.com/landing/acts [Accessed 10 Oct 2020].

---

## [Decision Letter · Decision Letter 1]

23 Oct 2020

High global consumption of potentially inappropriate fixed dose combination antibiotics: analysis of data from 75 countries

PONE-D-20-01816R1

Dear Dr. Hsia,

We’re pleased to inform you that your manuscript has been judged scientifically suitable for publication and will be formally accepted for publication once it meets all outstanding technical requirements.

Kind regards,

Iddya Karunasagar

Academic Editor

PLOS ONE

Additional Editor Comments (optional):

All reviewer comments have been addressed.

Reviewers' comments:

Reviewer's Responses to Questions

**Comments to the Author**

1. If the authors have adequately addressed your comments raised in a previous round of review and you feel that this manuscript is now acceptable for publication, you may indicate that here to bypass the “Comments to the Author” section, enter your conflict of interest statement in the “Confidential to Editor” section, and submit your "Accept" recommendation.

Reviewer #2: All comments have been addressed

2. Is the manuscript technically sound, and do the data support the conclusions?

Reviewer #2: Yes

3. Has the statistical analysis been performed appropriately and rigorously? 

Reviewer #2: Yes

4. Have the authors made all data underlying the findings in their manuscript fully available?

Reviewer #2: Yes

5. Is the manuscript presented in an intelligible fashion and written in standard English?

Reviewer #2: Yes

6. Review Comments to the Author

Reviewer #2: Thank you for revising the manuscript. The authors have addressing all the comments from the reviewer.

7. PLOS authors have the option to publish the peer review history of their article (what does this mean?). If published, this will include your full peer review and any attached files.

Reviewer #2: No

---

## [Editor Report · Acceptance letter]

2 Dec 2020

PONE-D-20-01816R1 

High global consumption of potentially inappropriate fixed dose combination antibiotics: analysis of data from 75 countries 

Dear Dr. Hsia:

I'm pleased to inform you that your manuscript has been deemed suitable for publication in PLOS ONE. Congratulations! Your manuscript is now with our production department. 

Kind regards, 

on behalf of

Dr. Iddya Karunasagar 

Academic Editor

PLOS ONE